# Unraveling the Impact of Acetylation Patterns in Chitosan Oligomers on Cu^2+^ Ion Binding: Insights from DFT Calculations

**DOI:** 10.3390/ijms241813792

**Published:** 2023-09-07

**Authors:** Ratna Singh, Jens Smiatek, Bruno M. Moerschbacher

**Affiliations:** 1Institute for Biology and Biotechnology of Plants, University of Münster, Schlossplatz 8, 48143 Münster, Germany; 2Institute for Computational Physics, Universität Stuttgart, Allmandring 3, 70569 Stuttgart, Germany; smiatek@icp.uni-stuttgart.de

**Keywords:** *N*-acetyl-d-glucosamine (A), d-glucosamine (D), copper-chitosan complex, GlcN subunit with Cu^2+^ bound at 2′NH_2_ (D-Cu = M), quantum mechanics, density functional theory (DFT), frontier molecular orbital theory (FMO), average local ionization energy (Ī)

## Abstract

Chitosans are partially acetylated polymers of glucosamine, structurally characterized by their degree of polymerization as well as their fraction and pattern of acetylation. These parameters strongly influence the physico-chemical properties and biological activities of chitosans, but structure-function relationships are only poorly understood. As an example, we here investigated the influence of acetylation on chitosan-copper complexation using density functional theory. We investigated the electronic structures of completely deacetylated and partially acetylated chitosan oligomers and their copper-bound complexes. Frontier molecular orbital theory revealed bonding orbitals for electrophiles and antibonding orbitals for nucleophiles in fully deacetylated glucosamine oligomers, while partially acetylated oligomers displayed bonding orbitals for both electrophiles and nucleophiles. Our calculations showed that the presence of an acetylated subunit in a chitosan oligomer affects the structural and the electronic properties of the oligomer by generating new intramolecular interactions with the free amino group of neighboring deacetylated subunits, thereby influencing its polarity. Furthermore, the band gap energy calculated from the fully and partially deacetylated oligomers indicates that the mobility of electrons in partially acetylated chitosan oligomers is higher than in fully deacetylated oligomers. In addition, fully deacetylated oligomers form more stable complexes with higher bond dissociation energies with copper than partially acetylated ones. Interestingly, in partially acetylated oligomers, the strength of copper binding was found to be dependent on the pattern of acetylation. Our study provides first insight into the influence of patterns of acetylation on the electronic and ion binding properties of chitosans. Depending on the intended application, the obtained results can serve as a guide for the selection of the optimal chitosan for a specific purpose.

## 1. Introduction

Inorganic copper compounds are not only important materials used in electronics, but they also have various biological applications such as in agriculture as fungicidal [1] and bactericidal agents, or in medicine as antiulcer or analgesic treatments [2]. Similarly, chitosan biopolymers, i.e., partially de-*N*-acetylated derivatives of chitin, have interesting material properties, but they are also considered to be promising functional biopolymers with multiple bioactivities such as antimicrobial activities, immuno-stimulatory activities in humans, animals, and plants, as well as wound healing and plant growth promoting activities [3]. Interestingly, chitosans strongly interact with metal ions, including copper ions. Therefore, chitosans can be used as metal absorbents, e.g., in waste management or water purification. Chitosan complexed with transition metals also finds application in building a matrix for enzyme immobilization [3]. Finally, copper-chitosan complexes can synergistically combine the antimicrobial properties of both ingredients [4].

Understanding the interaction of metal ions with chitosan oligomers or polymers at a molecular level is a prerequisite for developing and optimizing applications based on chitosan-metal complexes. Chitosans are linear co-polymers of β-1,4-linked *N*-acetyl-d-glucosamine (GlcNAc, A) and d-glucosamine (GlcN, D) units. Thus, they differ in their degrees of polymerization (DP), fractions of acetylation (*F*A), and patterns of acetylation (PA). Both the physico-chemical properties and the biological functionalities of chitosans strongly depend on these structural parameters [5]. Hence, for the development of successful applications based on the metal-complexing abilities of chitosans, it is crucial to understand the influence of *F*A and PA on their own structure as well as on copper-chitosan coordination. As the binding of metal ions to chitosans is conveyed by the free electron pair of the nitrogen in the GlcN unit, the influence of *F*A on copper binding is obvious. Previous studies on metal uptake by chitosan [6] have focused on copper-chitosan coordination geometry [7,8], but to our knowledge, no data is available regarding a possible influence of PA on copper coordination.

In the present study, we analyzed the electronic properties of completely deacetylated and partially acetylated chitosan oligomers and quantitatively characterized their ability to form complexes with copper. The analysis consisted of four steps. Each step increased in complexity by increasing the chain length of the chitosan oligomers studied:
i  Electronic structure of the GlcN monomer (D) and its copper-bound complex (D-Cu = M) were investigated regarding its coordination and assessing the stability of bridge and pendant model.ii GlcN- and GlcNAc (A)-containing dimers (D-D, A-D, D-A) were studied to analyze the effect of reducing and non-reducing ends on copper binding (M-D, D-M, A-M, M-A).iiiChitosan trimers (D-D-D, A-D-D, D-D-A, A-D-A) were analyzed to define the combined effect of reducing and non-reducing ends on the metal binding subunit (M-D-D, D-M-D, D-D-M, A-M-D, D-M-A, A-M-A).ivChitosan pentamers (D-A-D-D-D, D-D-D-A-D, D-A-D-A-D) were investigated to define the influence of the pattern of acetylation, i.e., of neighboring deacetylated or acetylated units, on copper binding (D-A-M-D-D, D-D-M-A-D, D-A-M-A-D) without the effects of reducing and non-reducing ends.

The results of the current study clearly show that PA has a significant influence on the electronic properties of chitosans and, correspondingly, on the formation and stability of copper-chitosan complexes.

## 2. Results and Discussion

To investigate the effect of PA within chitosan chains on copper binding, the structures of chito-oligomers and copper-bound chitosan oligomers were studied using density functional theory (DFT). The electronic structures of the monomer and oligomers were analyzed based on their wave functions to quantify the local polarizability, electron density at the molecules’ surface, and the molecular orbitals with their corresponding energies.

Considering the free energy and enthalpy of individual components and their complexes [9], the stabilities of the complexes were defined in terms of free energy of complex formation and bond dissociation energy. Further, the feasibility of electron mobility in the chito-oligomers was defined in terms of band gap energies.

### 2.1. Monomer

To determine the electronic properties of the GlcN monomer, the electrostatic potential (EPS) and the EPS surface extreme were mapped on the van der Waals (VDW) surface of GlcN monomer (D). The analysis revealed the position of the global minimum at the 1′O site (−49.6 kcal/mol), while the global maximum is at the 6′H group (62.1 kcal/mol) (Figure 1), indicating the most electropositive and the most electronegative sites on GlcN. Further, to map the ideal electrophilic reaction site, lowest average local ionization energies (Ī) were identified on GlcN to define the most vulnerable site to electrophilic attack. As expected, the lowest Ī in GlcN was mapped on the deprotonated amino group, indicating the position where electrons are least tightly bound and, thus, denoting the most probable site for the electrophilic copper attack.

For the copper-chitosan coordination, two different models have been proposed in the literature, i.e., bridge and pendant. It has been described that in the bridge model, the amino groups of two GlcN units coordinate with copper via intra- or intermolecular complexation within the same chain or between adjacent chains. In contrast, in the pendant model, copper coordinates with a single amino group. We examined both models using quantum chemical methods to define the most stable complex corresponding to their coordination geometry.

To this end, different coordination geometries of bridge and pendant forms of the GlcN-Cu complex were generated. For building the bridge geometry, copper was placed between 2′NH_2_ and 3′OH, surrounded by two, three, or four water molecules to generate tetra-, penta-, and hexa-coordination state geometries. The selection of 2′NH_2_ and 3′OH was based on the fact that in the later study, we wanted to observe the effect of neighboring subunits on the copper binding subunit; thus, 2′NH_2_ and 3′OH were identified as the most appropriate sites for coordination. For DFT calculations, generalized gradient approximation (GGA) was used by applying the Becke-Lee-Yang-Parr (BLYP) function.

In the geometry-optimized structure of copper-bound GlcN monomer, the bond length measured between Cu and 2′NH_2_ was 2.0 Å in both the bridge and pendant models, independent of the coordination geometries (Appendix A). In contrast, the bond length between Cu and 3′OH in the bridge model increased from 2.0 Å to 2.3 Å with increasing coordination number from tetra- to hexa. Further, based on calculations on the bound and unbound states of the complexes and their components, the free energies of complex formation and bond dissociation energies were analyzed. The results of the calculations are summarized in Appendix A, showing that increasing the copper coordination number influences the free energy of complex formation and bond dissociation energy in a steady way. This effect was seen in both the bridge and pendant models. From tetra- to hexa-coordination, the free energy of complex formation increased (became less negative) from ca. −69.9 kcal/mol to ca. −20.9 kcal/mol in the bridge model and from ca. −36.7 kcal/mol to ca. −12.9 kcal/mol in the pendant model. Concomitantly, the bond dissociation energy decreased from ca. 85.7 kcal/mol to ca. 30.8 kcal/mol in the bridge model and from ca. 52.7 kcal/mol to ca. 28.7 kcal/mol in the pendant model. The coordination of copper with GlcN is discussed controversially in the literature. Domard et al. identified the pendant complex as more stable than the bridge complex [8], whereas Lü et al. reported that the bridge complex is more stable than the pendant one [7]. Our results based on calculating free energies of complex formation and bond dissociation energies suggest that for the GlcN monomer, the bridge model is better than the pendant model and that tetra-coordination in the monomer is more stable than penta- and hexa-coordination.

### 2.2. Dimers

For identifying the reactive sites in dimers, the electronic structure of the fully deacetylated dimer GlcN-GlcN (D-D) and of the two possible partially acetylated dimers GlcNAc-GlcN (A-D) and GlcN-GlcNAc (D-A) were examined. Like in the GlcN monomer, the lowest ionization energy (Ī) in the dimers was mapped at the 2′NH_2_ group in all the dimers. In D-D, the 2′NH_2_ of the reducing end subunit was identified as the most susceptible site for copper attack (Figure 2a). These results were corroborated by frontier orbital analysis (Figure 3), where HOMO mapped on the 2′NH_2_ of the reducing end subunit of D-D. Interestingly, from the calculation, we observed that the reactivity of the free amino group of both GlcN units was strongly influenced by the neighboring unit, as the ionization energy dropped in both cases when it was neighbored by an acetylated GlcNAc unit. At the non-reducing end, it decreased from 6.17 eV in D-D to 6.05 eV in D-A, while at the reducing end, it decreased from 6.03 eV in D-D to 5.96 eV in A-D (Figure 2a–c). As shown in Figure 2c, the acetylated GlcNAc unit at the reducing end of the dimer D-A leads to a higher positive charge on the reducing end’s 1′OH group compared to the dimers D-D and A-D carrying a deacetylated GlcN unit at the reducing end. To analyze this more quantitatively, EPS global maxima were mapped at the reducing end’s 1′OH, where the electrostatic potential energy on 1′H was 49.0 kcal/mol in D-D and 55.2 kcal/mol in A-D, but reached 68.2 kcal/mol in D-A. Similarly, the potential energy measured at 1′O was −29.3 kcal/mol in D-D, −30.1 kcal/mol in A-D, and −12.3 kcal/mol in D-A. Thus, the additive energy calculated from 1′O and 1′H showed a difference of 36.2 kcal/mol between D-D and D-A, implying that the acetyl group at the reducing end delocalizes the electron density and affects the charges on it. Therefore, combining ionization energy with EPS results suggests that in chitosan dimers, PA significantly impacts electronic properties and, presumably, their pKa and metal ion binding capability.

To analyze the relative stability of the copper-bonded complexes in their ground states, calculations were performed on these structures. Starting from the fully deacetylated dimer (D-D), two copper-bonded complexes were generated in bridged form, where the copper was placed between 2′NH_2_ and 3′OH of the non-reducing end of the GlcN unit (M-D) and coordinated with water to reach the coordination state, and the same step was performed at the reducing end of the GlcN unit (D-M). Moreover, all possible coordination states (tetra-, penta-, and hexa-coordination) with deacetylated and partially acetylated dimers bound with copper were generated in the bridge model, and their structures were geometrically optimized using the BLYP method. Unexpectedly, only for the hexa-coordination state, the copper-bound geometries (D-M, M-D, A-M, and M-A) were successfully converged, while for the tetra- and penta-coordination’s, geometry optimization failed, possibly suggesting that in oligomers, unlike in the GlcN monomer, hexa-coordination is the most reasonable geometry and preferred over tetra-coordination. Accordingly, complex formation free energies and bond dissociation energies were calculated for hexa-coordination states. The calculations indicated that copper bound at the reducing end GlcN unit in D-M has a lower free energy of complex formation and a higher bond dissociation energy (Figure 4a) than copper bound at the non-reducing end GlcN unit in M-D. This difference in stability may be explained based on the difference in the ionization energies on the –NH_2_ group of the reducing and non-reducing end subunit (Figure 2a) as well as by the intramolecular interaction occurring in the D-M complex, between a copper-bound water and the 6′OH of the non-reducing end unit (Appendix A). Moreover, the difference in bond dissociation energy can also be partially explained by the bond length between Cu and 3′OH (Appendix A). While the bond length between Cu and 2′NH_2_ was 2.0 Å in both dimers, the bond length between Cu and 3′OH was 2.1 Å in D-M and 2.3 Å in M-D, the increased bond length indicating a weaker interaction, thus explaining the lower bond dissociation energy in M-D. Taken together, these results suggest that a copper-complex at the non-reducing end GlcN dissociates more easily than a copper bound at the reducing end GlcN.

To analyze a possible effect of the acetylation state of the neighboring subunit on the stability and the binding energy of the complex, calculations were also carried out on A-M and M-A complexes. Comparing the free energies of complex formation as well as the bond dissociation energies (Figure 4) of M-D with M-A, and of D-M with A-M, indicated that the presence of an acetylated subunit at either end of the dimer influences the stability of the copper complexes. The free energy of complex formation was higher (less negative) when the copper-bound GlcN-unit was neighbored by an acetylated GlcNAc unit, especially when the GlcNAc unit was at the non-reducing end of the dimer (D-M versus A-M), suggesting destabilization of the complex by acetylation of the neighboring units. Irrespective of whether the copper-binding GlcN unit was neighbored by a deacetylated GlcN unit (D-M and M-D) or by an acetylated GlcNAc unit (A-M and M-A), copper bound at the reducing end was more stable than copper bound at the non-reducing end. The preference for copper binding at the reducing end over the non-reducing end GlcN unit in the fully deacetylated dimer as well as the destabilizing effect of a neighboring acetylated GlcNAc unit were corroborated by calculations of the bond dissociation energies which were higher in D-M than in M-D, and higher in D-M than in A-M, but only slightly higher in M-D than in M-A (Appendix A).

Like in the fully deacetylated dimer D-D, the results obtained with the partially acetylated dimers A-D and D-A can also be explained by their ionization energies, an interaction with the neighboring subunit’s 6′OH, as well as a smaller 3′OH bonding distance to copper in the A-M complex compared to the M-A complex (Appendix A).

### 2.3. Trimers

When the local reactive behavior of the fully deacetylated GlcN trimer D-D-D was analyzed based on average local ionization energies, the 2′NH_2_ group at the reducing end (6.05 eV) subunit was identified as the most probable site for electrophilic attack (Figure 2), as also supported by frontier orbital analysis as shown in Figure 3. Like in the dimers, acetylation of the non-reducing end subunit (A-D-D) influenced the ionization energy of the neighboring, middle subunit’s 2′NH_2_, increasing from 6.07 eV in D-D-D to 6.95 eV in A-D-D. In contrast, acetylation of the reducing end (D-D-A) or at both ends (A-D-A) did not have a strong impact on the 2′NH_2_ group of the middle subunit. Structural analysis of DFT-based geometry-optimized trimers showed that the ionization energy of the 2′NH_2_ group of the middle subunit in A-D-D increased relative to D-D-D due to the bonding between the 2′N of the middle subunit and the 6′H of the reducing end subunit in A-D-D (Figure 2 and Appendix A). This interaction was not observed in the other trimers (Appendix A), suggesting that the acetylated subunit at the non-reducing end has an effect on the structural conformation of the complex, bringing the other two GlcN subunits closer to each other, facilitating the new bonding between 2′N and 6′H, and that influences its polarity. This analysis indicates that depending on its position, an acetylated subunit in an oligomer has an impact on its structure and, consequently, on its molecular properties. Moreover, EPS calculations on trimers (Figure 2) showed that an acetylated subunit at the reducing end (D-D-A and A-D-A) influences the charge distribution, decreasing the electron density and thus increasing the positive charge on 1′OH (Figure 2). The energy maxima and minima on 1′OH of the reducing end subunit showed that compared to D-D-D, the energy differences in D-D-A and A-D-A were almost 21.0 kcal/mol and 22.6 kcal/mol, respectively (Figure 2). In contrast, acetylation of the non-reducing end increases the negative charge on 3′OH, and more prominently so in A-D-D than in A-D-A. Overall, the results indicate that in terms of surface charge distribution, A-D-D is the least positively charged trimer compared to D-D-D, D-D-A, and A-D-A, again indicating that in small oligomers, the position of an acetylated group strongly influences the property of the oligomers. 

To analyze a possible influence of the acetylation state of the reducing and non-reducing end subunits on copper binding to the middle subunit and, correspondingly, on the stability of the copper-chitosan trimer complex, we first carried out DFT calculations on the completely deacetylated trimer D-D-D with and without copper bound to any of the GlcN units (M-D-D, D-M-D, and D-D-M) (Figure 4b). As for the dimers, the complex with copper bound to the reducing end subunit (D-D-M) had the lowest free energy of complex formation and, correspondingly, the bond dissociation energy measured between copper and the trimers was highest for D-D-M (Figure 4b). This also correlates with the average local ionization which measured lowest (6.05 eV) at the reducing end of D-D-D. With the BLYP method, geometry optimization failed for the complex where copper was bound at the non-reducing end of the fully deacetylated trimer (M-D-D) which possibly is due to high local ionization energy (6.21 eV) and higher positive charge on 2′NH_2_ of the non-reducing end subunit. Thus, to confirm the results, DFT calculations using the Perdew-Burke-Ernzerhof (PBE) function with dispersion correction were performed. Results obtained using the PBE function for dimer and trimer complexes aligned well (Appendix A) with those obtained using the BLYP function. The bond dissociation energy and the stability of the M-D-D complex was found to be the lowest compared to the other trimers.

Further, to map the effect of an acetylated subunit on the stability of the complex, calculations were carried out on the trimers D-D-D, A-D-D, D-D-A, and A-D-A, as well as their counterparts with copper bound at the middle GlcN subunit (D-M-D, A-M-D, D-M-A, and A-M-A). Calculations of the free energies of complex formation revealed a decreasing stability of the complexes in the sequence D-M-A > A-M-A > D-M-D > A-M-D, indicating that an acetylated subunit at the non-reducing end decreased the stability of the copper-chitosan complex, while it was increased by the presence of an acetylated subunit at the reducing end. The stabilizing influence of the reducing end acetylation was stronger than the destabilizing influence of the non-reducing end acetylation, as the double-acetylated trimer A-M-A was slightly more stable than the fully deacetylated trimer D-M-D. While D-M-A and D-D-M have similarly high free energy of complex formation, the bond strength in D-D-M (Appendix A) was stronger. Again, the differences in the bond dissociation energies can be explained based on the bond length between 3′OH and copper which measured 2.2 Å in D-M-A, whereas it was 2.4 Å in D-M-D and A-M-A, and 2.5 Å in A-M-D. These results suggest that not only the 2′NH_2_, but also the interaction of copper with the 3′OH has a significant effect on the stability of the copper-chitosan complexes.

### 2.4. Pentamer

To analyze the effect of an acetylated subunit within the chitosan chain (and excluding the effect of reducing and non-reducing end positions), calculations were carried out on pentamers, starting with the fully deacetylated one (D-D-D-D-D). As shown in Figure 2h, the lowest ionization energy was mapped on the deprotonated amino group of the middle-most GlcN unit (6.17 eV). To study the effect of a neighboring acetylated group on the ionization energy of the middle GlcN unit, an acetylated GlcNAc unit was placed on either or both sites of it, generating the following sequences: D-A-D-D-D, D-D-D-A-D, and D-A-D-A-D. In all three cases, the presence of an acetylated unit influenced the local nucleophilicity of 2′NH_2_ of the GlcN unit in the middle, increasing its ionization energy from 6.17 eV in the fully deacetylated pentamer strongly to ca. 7.32 eV in the partially acetylated ones. As a consequence of this strong increase, in the mono-acetylated pentamers (D-A-D-D-D and D-D-D-A-D), only three of the four amino groups mapped with lower ionization energy (Figure 2i,k), indicating three probable electrophilic attack sites and, thus, potential copper binding sites, and, consequently, only two instead of three in the double deacetylated pentamer (D-A-D-A-D, Figure 2j). The effect on the polarity of the amino group in partially acetylated oligomers can again be explained by intramolecular bonding (1.8 Å) between the 2′N of the middle unit with the 6′OH of the neighboring unit (to its ‘right’ side, i.e., towards the reducing end of the pentamer) (Appendix A). Interestingly, this interaction was observed only in the presence of one or two acetylated units next to the middle deacetylated one, and even independently of the position of the acetylated unit ‘left’ or ‘right’ of the middle unit, which indicates that an acetylated GlcNAc unit affects the structural geometry of the chitosan chain due to intramolecular interactions, and thus influences the polarity of the amino groups. Hence, it follows that in a chitosan chain, the presence of an acetylated GlcNAc unit affects the nucleophilicity of neighboring free amino groups and, thus, their accessibility for electrophilic attack and copper binding. The EPS maps of the completely deacetylated and the partially acetylated oligomers also confirmed the above analysis and showed that in the oligomer (and, consequently, also in polymers), the surface charge on the amino group of a deacetylated GlcN unit decreases (Figure 2h–k) in the presence of an acetylated GlcNAc unit adjacent to it.

Finally, copper-chitosan pentamer complexes were generated in which the copper was bound to the middle GlcN unit, i.e., D-D-M-D-D, D-A-M-D-D, D-A-M-A-D, and D-D-M-A-D. Calculations were carried out on geometry-optimized structures by using the BLYP method. In all pentameric complexes, the 3′OH-Cu and 2′NH_2_-Cu bond lengths were measured as 2.1 Å and 2.0 Å, respectively.

Results obtained from the calculation of free energies of complex formation and the bond dissociation energies are shown in Figure 5 and Table 1. As expected from the ionization energies described above, a neighboring GlcNAc unit strongly increased the free energies of complex formation and also slightly decreased the bond dissociation energies. However, the effect was not strongly influenced by the position of the acetylated GlcNAc unit ‘right’ or ‘left’ of the copper bound GlcN unit.

### 2.5. Energy Gap

In frontier molecular orbital (FMO) analysis, we examined the chemical reactivity of completely deacetylated and partially acetylated oligomers by mapping the position and the energies of the highest occupied molecular orbital (HOMO) and the lowest unoccupied molecular orbital (LUMO). For the partially acetylated dimers, LUMO was mapped on the acetylated group while HOMO was located on the NH_2_ group of the deacetylated subunit (Figure 3), and the same was observed for partially acetylated trimers and pentamers (Figure 3 and Appendix A). As HOMO and LUMO indicate the most likely sites for electrophilic and nucleophilic attack, respectively, this suggests that the acetamido groups in partially acetylated oligomers are accessible for nucleophilic reactions, and the deprotonated free amino groups for electrophilic reactions. Our calculations on fully deacetylated, deprotonated oligomers (D-D, D-D-D, D-D-D-D, D-D-D-D-D) indicate positive LUMO energies at the nonreducing end subunit (Table 1), suggesting that the non-reducing end of deacetylated oligomers possess anti-bonding orbitals and, consequently, exhibit negative electron affinities. However, on partially acetylated oligomers, negative HOMO and LUMO energies indicate the presence of bonding orbitals for both electrophiles and nucleophiles, although the lower HOMO energies in monoacetylated oligomers also suggest their weak interactions with electrophiles. The lower band gap energies (Table 1) calculated from the ground state of acetylated oligomers suggest that electron transfer within the molecule from HOMO to LUMO is easier in acetylated oligomers (due to negative LUMO energies) than in deacetylated oligomers. Notably, compared to the smaller oligomers (DP_2-4_), higher HOMO energies were mapped in the pentamer (−4.98 eV) and hexamer (−4.97 eV), indicating their higher reactivity towards electrophiles (Table 1). Increased HOMO energies in pentamer and hexamer compared to smaller oligomers possibly indicate the minimum length of oligomers required for efficient copper binding. 

These results are consistent with the above-described free energy calculations and explain why completely deacetylated oligomers form more stable complexes (due to their high HOMO energies) than partially acetylated ones. 

## 3. Materials and Methods

Monomer, dimer, and trimer structures of chitosan were built using Avogadro [10], and geometry optimization was carried out using MMFF94s [11], force field. To build the bridge geometry, copper was placed between 2′NH2 and 3′OH, surrounded by two, three, or four water molecules to generate tetra-, penta-, and hexa-coordination state geometries. Similarly, for the pendant model, copper was placed near the 2′NH2, and tetra-, penta-, and hexa-coordination geometries were generated by adding three, four, or five water molecules. DFT calculations were performed with the ORCA [12] software using GGA [13,14]—based wave functions and the Perdew-Burke-Ernzerhof (PBE) and Becke-LYP (BLYP) methods [15] with def2-TZVPP basis set [16,17]. In PBE, for dispersion correction, the Becke-Johnson damping (D3BJ) method [18] was applied. To incorporate the effect of the solvent phase, the dielectric constant of water was used, and geometry optimization was carried out with slow convergence. Bond lengths, HOMO-LUMO energies and their excitation energies, free energy, entropy, etc., were deduced from the optimized structures. In DFT calculations, internal electronic energies were converted to Gibbs free energies by including thermal corrections arising from translational, rotational, and vibrational nuclear motions, together with entropies using the following formulae.

The inner energy is:

U = E(el) + E(ZPE) + E(vib) + E(rot) + E(trans)
(1)

where E(el) is the total energy from the electronic structure, E(ZPE) is the zero-temperature vibrational energy from the frequency calculation, E(vib) is the temperature correction to E(ZPE), and E(rot) and E(trans) are the rotational and translational thermal energies, respectively.

The enthalpy is:H = U + kB × T(2)
where kB is Boltzmann’s constant.

The entropy contributions are:T × S = T × (S(el) + S(vib) + S(rot) + S(trans))(3)
where T is temperature, S is entropy, and S(el), S(vib), S(rot), and S(trans) are electronic, vibrational, rotational, and translational entropies, respectively.

The Gibbs free enthalpy is:

G = H − T × S
(4)

where H is enthalpy, T is temperature, and S is entropy.

Based on the free energy and enthalpy of individual components (G_A_, G_B_, H_A_, H_B_) and their complexes (G_AB_, H_AB_), free energy of complex formation and bond dissociation energies were calculated for the copper-bound chitosan oligomers in completely deacetylated and partially acetylated forms at 298.15 K, based on the following formulae:Free energy of complex formation = G_AB_ − (G_A_ + G_B_) (5)
Bond dissociation energy = H_A_ + H_B_ − H_(AB)_(6)
where the formation energy of the complexes was evaluated in terms of Gibbs free energy (G) and the bond dissociation energy was calculated from the enthalpy (H).

Multiwfn 3.4.1 [19] was used to analyze the average local ionization energy using the formula:(7)$$I^{-}=\sum_{i}\rho i(r)|\varepsilon i\;|\;÷\;\rho (r)$$
where εi  and ρi(r) are the orbital energy of the *i*th molecular orbital and the electron density.

Molecular electrostatic potential (ESP) was calculated by taking the 0.001 isosurface of electron density on vdW surface. The position of maxima and minima analyzed on the vdW surface using Multiwfn 3.4.1. Further, Molden 2.0 [20] and Avogadro were applied to study the HOMO and LUMO sites in oligomers. All the figures were generated with VMD [21] and USCF chimera [22].

## 4. Conclusions

Our study shows that in small chitosan oligomers, the presence of an acetylated GlcNAc or a deacetylated GlcN unit at the reducing and/or nonreducing end of the oligomers has a strong influence on the electronic structure of the oligomers and, as a consequence, on their chemical properties. We showed that in a partially acetylated chitosan, the sequence of GlcN and GlcNAc units, i.e., the pattern of acetylation, has an effect on the polarity of the free amino groups which is influenced by the acetylation state of its neighboring units because of intramolecular interactions between the subunits. FMO analysis based on HOMO energies revealed that fully deacetylated penta- and hexamers exert higher reactivity towards electrophiles than smaller oligomers. The lower energy gap in partially acetylated oligomers suggests that mobility of electrons is more feasible in partially acetylated oligomers than in completely deacetylated ones. 

The study of copper-chitosan complexes showed that the completely deacetylated oligomers formed more stable copper complexes with higher (less negative) dissociation energies than the partially acetylated oligomers. Thus, we conclude that in partially acetylated chitosans, the copper loading and copper release ability can be controlled by both the number and position of acetylated and deacetylated units, i.e., by the *F*A and the PA of the chitosan.

## Figures and Tables

**Figure 1 ijms-24-13792-f001:**
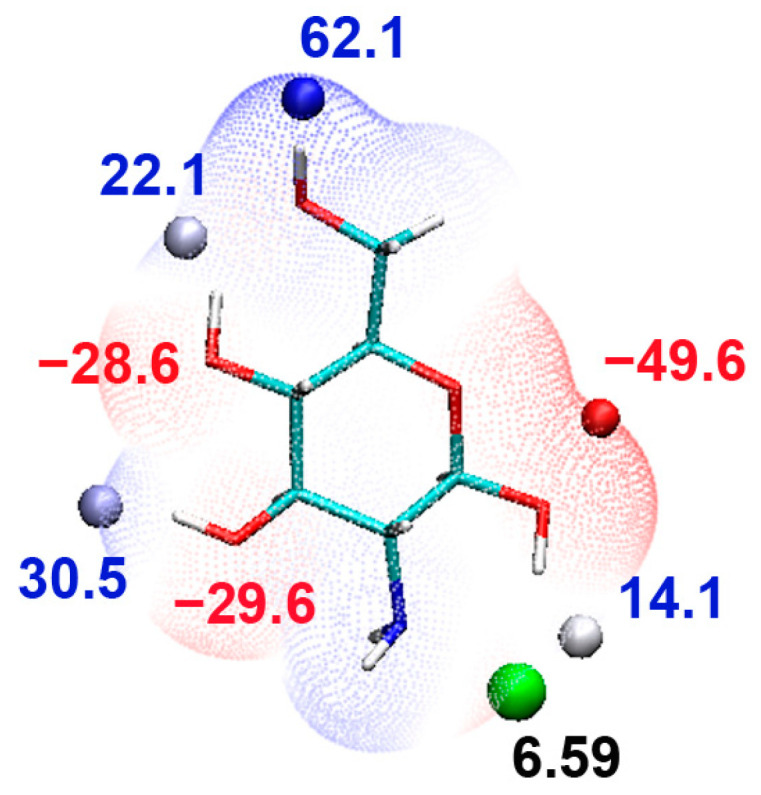
Monomer with the lowest local ionization (eV) values (green dots, black labels) at the 2′NH2 site indicating most probable site for electrophilic attack. Electrostatic surface charge distribution shown on the surface of monomer at the scale bar −72 to 72 kcal/mol. Extrema with their global minima and maxima are displayed at 1′OH (red label) and 6′OH (blue label) surface of glucosamine.

**Figure 2 ijms-24-13792-f002:**
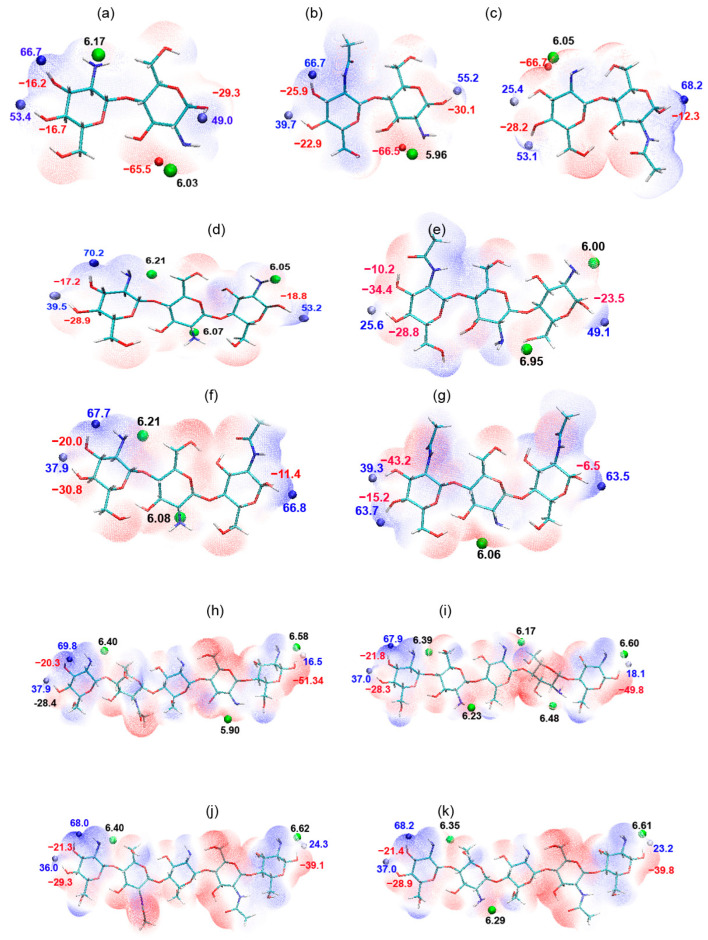
Chitosan oligomers with their local ionization (eV) energies (green dots, black labels), lower value displaying more preferred site for electrophilic attack. Electrostatic surface charge distribution showed on the surface of oligomers at the scale bar −72 to 72 kcal/mol, where minima and maxima energy value displayed on reducing and non-reducing end of 1ÒH, 3ÒH, and 4ÒH, labels in red are for negative charge and in blue are for positive charge; (**a**) GlcN-GlcN (D-D); (**b**) GlcNAc-GlcN (AD); (**c**) GlcN-GlcNAc (D-A); (**d**) GlcN-GlcN-GlcN (D-D-D); (**e**) GlcNAc-GlcN-GlcN (A-D-D); (**f**) GlcN-GlcN-GlcNAc (D-D-A); (**g**) GlcNAc-GlcN-GlcNAc (A-D-A); (**h**) GlcN-GlcN-GlcN-GlcN-GlcN (D-D-D-D-D); (**i**) GlcN-GlcNAc-GlcN-GlcN-GlcN (D-A-D-D-D); (**j**) GlcN-GlcNAc-GlcN-GlcNAc-GlcN (D-A-D-A-D); (**k**) GlcN-GlcN-GlcN-GlcNAc-GlcN (D-D-D-A-D).

**Figure 3 ijms-24-13792-f003:**
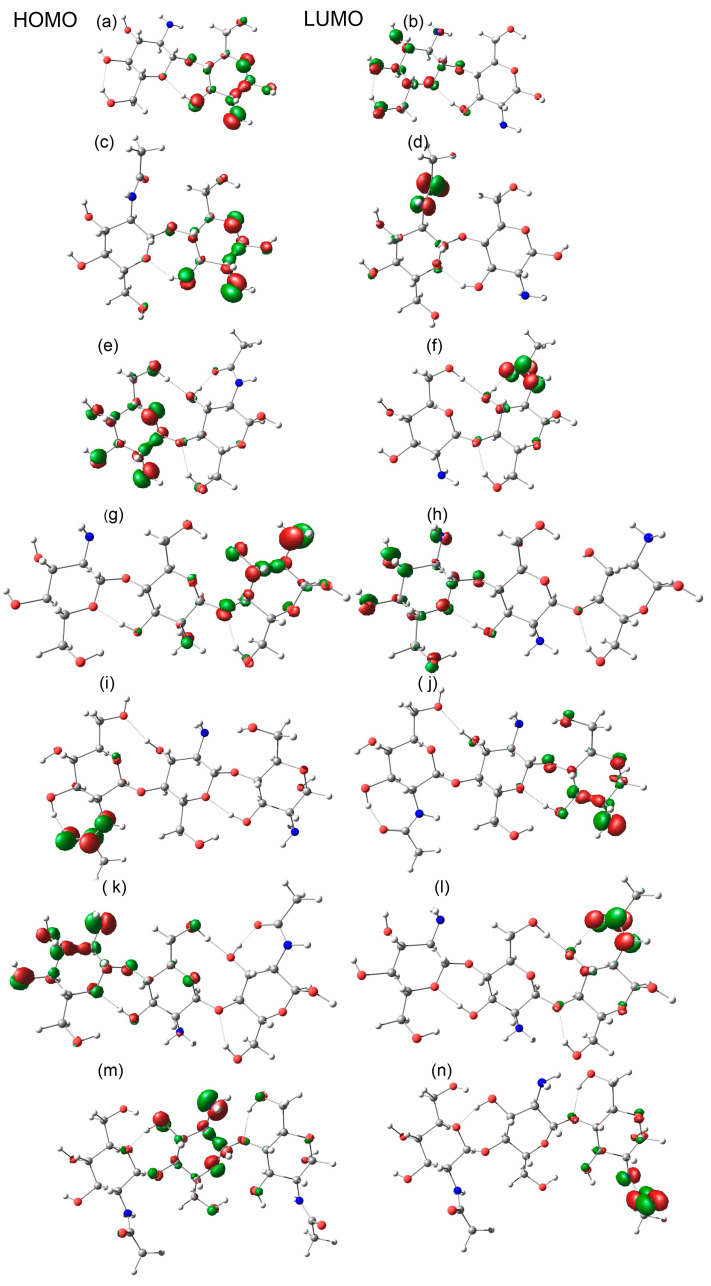
HOMO and LUMO orbitals position are showed on chito dimers and trimers (**a**) GlcN-GlcN (D-D), HOMO (**b**) GlcN-GlcN (D-D), LUMO (**c**) GlcNAc-GlcN (A-D), HOMO (**d**) GlcNAc-GlcN (A-D), LUMO (**e**) GlcN-GlcNAc (D-A), HOMO (**f**) GlcN-GlcNAc (D-A), LUMO (**g**) GlcN-GlcN-GlcN (D-D-D), HOMO (**h**) GlcN-GlcN-GlcN (D-D-D), LUMO (**i**) GlcN-GlcN-GlcNAc (D-D-A), HOMO (**j**) GlcN-GlcN-GlcNAc (D-D-A), LUMO (**k**) GlcNAc-GlcN-GlcN (A-D-D), HOMO (**l**) GlcNAc-GlcN-GlcN (A-D-D), LUMO (**m**) GlcNAc-GlcN-GlcNAc (A-D-A), HOMO (**n**) GlcNAc-GlcN-GlcNAc (A-D-A), LUMO.

**Figure 4 ijms-24-13792-f004:**
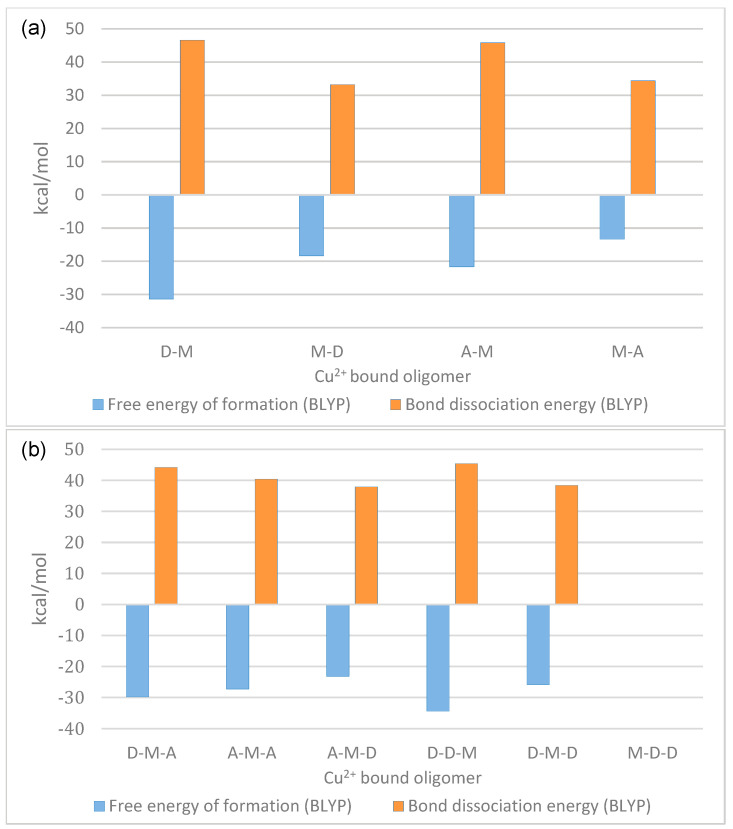
Free energies and bond dissociation energies of copper-chitosan complex (**a**) chitosan dimers (**b**) selected chitosan trimers, where M represents the Cu^2+^ bound to the 2′NH2 and 3′OH sites of the same subunit.

**Figure 5 ijms-24-13792-f005:**
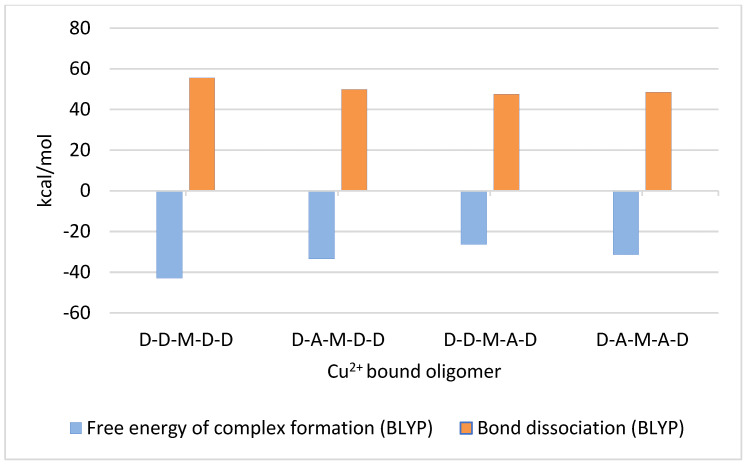
Free energies and bond dissociation energies of copper-chitosan complex formation in the fully deacetylated chitosan pentamer GlcN-GlcN-GlcN-GlcN-GlcN (D-D-D-D-D) and in the partially acetylated pentamers GlcN-GlcNAc-GlcN-GlcN-GlcN (D-A-D-D-D), GlcN-GlcN-GlcN-GlcNAc-GlcN (D-D-D-A-D), and GlcN-GlcNAc-GlcN-GlcNAc-GlcN (D-A-D-A-D), calculated using the BLYP method, where M represents the Cu^2+^ bound to the 2′NH2 and 3′OH sites of the same subunit.

**Table 1 ijms-24-13792-t001:** Band gap calculated from HOMO and LUMO energies.

Oligomer	HOMO (eV)	LUMO (eV)	Bandgap (eV)
**D**	−5.324	0.498	5.823
**DD**	−5.040	0.339	5.387
**DA**	−5.049	−0.688	4.355
**AD**	−5.039	−0.376	4.662
**DDD**	−5.173	0.413	5.587
**ADD**	−5.193	−0.621	4.572
**DDA**	−5.210	−0.685	4.473
**ADA**	−5.103	−0.439	4.663
**DDDD**	−5.229	0.4111	5.641
**DDDDD**	−4.986	0.415	5.401
**DADAD**	−4.898	−0.294	4.682
**DDDAD**	−5.185	−0.527	4.658
**DADDD**	−5.202	−0.519	4.603
**DDDDDD**	−4.974	0.482	5.457

## Data Availability

Geometry-optimized structures in pdb format are publicly available with Appendix A.

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
