# Peer review of "Unraveling the Impact of Acetylation Patterns in Chitosan Oligomers on Cu^2+^ Ion Binding: Insights from DFT Calculations"

_ijms, 2023, doi:10.3390/ijms241813792_

Round 1

Reviewer 1 Report

The article under consideration explores the interaction of copper with chitosan oligomers, specifically considering hexamers with acetylated and deacetylated units as monomers. The study emphasizes the significant impact of the reducing and non-reducing ends of the oligomers on their chemical properties. The main finding is that completely deacetylated oligomers form more stable complexes with Cu(II), as established through the analysis of free energy of complexation and FMO (Frontier Molecular Orbital) analysis. The study is presented systematically and in a well-organized manner.

However, there are a few concerns that need to be addressed by the authors before the article is ready for publication:

  1. The abstract is excessively long and should be made more concise.
  2. There are concerns regarding the choice of the DFT (Density Functional Theory) functional in this study. The authors should investigate whether the same conclusions hold true when using a hybrid functional, such as PBE0, B3LYP, or a meta-GGA functional in the M06 family. a single-point calculation on the optimized geometry of PBE/BLYP with a certain % of HF-exchange should be included for validation purposes. This will strengthen the reliability of the findings and provide a more comprehensive assessment of the interaction between copper and chitosan oligomers.
  3. The authors are encouraged to include the optimized coordinates in the Supplementary Information (SI) to facilitate reproducibility and allow other researchers to reuse and validate the results obtained in this study.

Once these issues are addressed, the manuscript will be ready for publication in IJMS.

Author Response

Response to Reviewer -1 (please consider the attached docx file for graphs)

  1. The abstract is excessively long and should be made more concise

Answer: As suggested, the abstract is now shorter

  1. There are concerns regarding the choice of the DFT (Density Functional Theory) functional in this study. The authors should investigate whether the same conclusions hold true when using a hybrid functional, such as PBE0, B3LYP, or a meta-GGA functional in the M06 family. a single-point calculation on the optimized geometry of PBE/BLYP with a certain % of HF-exchange should be included for validation purposes. This will strengthen the reliability of the findings and provide a more comprehensive assessment of the interaction between copper and chitosan oligomers.

Answer: As suggested, we performed single-point calculation using B3LYP function (with 25% of HF) on the geometry optimized structure of dimers (DD, AD, DA, MD, DM, MA, AM) from BLYP function. The graph below shows the comparison between PBE (Fig. S3), BLYP (Figure 3) and B3LYP, and as expected, we obtained similar values for the free energy of complex formation and the bond dissociation energy.

DFT calculation for dimers with PBE0/B3LYP was computationally very expensive, it takes 3-4 days for each complex. Thus, performing this long calculation again on trimers and pentamers and their complexes (DDD, ADD, DDA, ADA, MDD, DMD, DDM, AMD, DMA, AMA, DDDDD, DADDD, DDDAD, DADAD, DDMDD, DAMDD, DDMAD, DAMAD) will take months on HPC batch system (queue) and judged by the results obtained for the dimers, would provide the same information in the end.

Thus, we hope that the results obtained from dimers will satisfy the reviewer’s concerns regarding the choice of DFT functional theory.

  1. The authors are encouraged to include the optimized coordinates in the Supplementary Information (SI) to facilitate reproducibility and allow other researchers to reuse and validate the results obtained in this study.

Answer: All the geometry optimized structures are provided in a zip folder namedgeometry_optimized_structures. Structures optimized using BLYP function are in BLYP folders and structures optimized using PBE function are in PBE folders. Each pdb structure is named respective to their corresponding pattern containing acetylated (A) and deacetylated (D) subunits.

Reviewer 2 Report

The authors have investigated the electronic structures of completely deacetylated and partially acetylated chitosan oligomers (monomer, dimers, trimers, and pentamers) in the deprotonated states and their copper-bound complexes using density functional theory. Authors reported that fully deacetylated oligomers form more stable complexes with higher bond dissociation energies with copper than partially acetylated ones. This study provides insight into the influence of patterns of acetylation on the electronic and ion binding properties of chitosans. Overall, it is a good article and I recommend this to be published after a few minor revisions.

1.     Authors have not reported the optimized structure of the studied systems. I recommend providing optimized geometry with optimized parameters.

2. lines 437-451 need some formatting corrections. 

Author Response

  1. Authors have not reported the optimized structure of the studied systems. I recommend providing optimized geometry with optimized parameters.

Answer: All the geometry optimized structures are provided in a zip folder named geometry_optimized_structures. Structures optimized using BLYP function are in BLYP folders and structures optimized using PBE function are in PBE folders. Each pdb structure is named respective to their corresponding pattern containing acetylated (A) and deacetylated (D) subunits.

  1. lines 437-451 need some formatting corrections.

Answer: Thank you for correction suggestion. As suggested lines 437-451 are reformatted.

Round 2

Reviewer 1 Report

The concerns raised by this reviewer have been appropriately addressed by the authors. Considering their response and the revisions made to the manuscript, I believe that the paper is suitable for publication in its current state.